# A New Species and Two New Records of the Lichen Genus *Fissurina* from China

**Kaijie Shi, Zefeng Jia and Xin Zhao \***

College of Life Sciences, Liaocheng University, Liaocheng 252059, China
* Correspondence: zhaoxin1@lcu.edu.cn

**Abstract:** The lichenized fungal genus *Fissurina* with mostly slit-like lirellae, belongs to Graphidaceae and is mainly distributed in tropical and subtropical regions. A total of 17 *Fissurina* species have been reported from China. During a survey of the lichen diversity of southern China, a new species *Fissurina wuyinensis* K.J. Shi, Z.F. Jia and X. Zhao, sp. nov. was found, which is characterized by a corticolous thallus without detected secondary substances, uncarbonized lirellae, and an exposed disc with pruina, muriform and amyloid ascospores. Furthermore, two new records of *F. pseudostromatica*, *F. subcomparimuralis* have been identified by morphological, anatomical, chemical and molecular studies. Phylogenetic analyses of three loci (ITS, nuLSU and mtSSU) supported the position of these species within *Fissurina*. Detailed morphological descriptions as well as high-resolution photographs of the morphology and anatomy of the three species are provided, as well as a comparison and discussion of the characteristics of similar species. The studied specimens were deposited in the Fungarium of the College of Life Sciences, Liaocheng University (LCUF).

**Keywords:** lichenized fungi; Ascomycota; Ostropales; taxonomy

## 1. Introduction

Graphidaceae is the largest family of tropical and subtropical crustose lichens. It is a symbiotic ecosystem composed of lichenized fungi of various genera and species of Chlorophyta Chlorococcaceae *Trebouxia* spp. and Trentepohliaceae *Trentepohlia* spp. from the plant kingdom. It belongs to Ascomycota, Lecanoromycetes, Ostropomycetidae and Ostropales. The currently accepted Graphidaceae includes 4 subfamilies, more than 80 genera, and a total of more than 2000 species [1–4].

The lichen genus *Fissurina* Fée belongs to Graphidaceae, and is characterized by a pale yellow-brown to olive green (rarely whitish), mostly smooth and glossy thallus, mostly slit-like lirellae and ovoid ascospores with a halo, transversely septate or muriform, amyloid to non-amyloid [5–7]. Fée established the genus with *Fissurina dumastii* Fée as the type species [8], it was subsumed within *Graphis* Adans. And *Graphina* Müll. Arg. [9–11]. In 1992, with the description of *Fissurina quadrispora* by Kalb and Hafellner, *Fissurina* was resurrected to the generic level [12], and in 1999, Staiger and Kalb redescribed *Fissurina* [13]. Staiger studied the combination of the phenotypic characteristics (ascomata characteristics, asci structure, ascospores characteristics, etc.) and genotype of Graphidaceae. She proposed a taxonomy of the family based on features other than spores combined with existing molecular data. Enter the embryonic phase of the combination of phenotype and genotype; a more natural classification of the family Graphidaceae was defined and generally accepted [14]. After that, Kalb et al., Archer, Lücking et al., etc., revised Graphidaceae [15–18], and so far the lirellate-morph Graphidaceae classification system of 24 genera has been generally recognized, and the genus level of *Fissurina* status tends to be more scientifically defined. Subsequent phylogenetic studies confirmed its phylogenetic position in the family Graphidaceae [19]. Phylogenetic studies indicate that *Fissurina* as currently defined is polyphyletic, and together with some taxa of *Dyplolabia* A. Massal., *Ocellularia* G. Mey. and

*Myriotrema* Fée (the only common characteristic is the astrothelioid ascospores) constitute a monophyletic group, which is clearly distinct from other taxa of the family Graphidaceae and included in Fissurinoideae [1,20–22]. Kraichak and Lücking used a temporal banding approach to conduct in-depth studies on several orders of the two main subclasses of Lecanoromycetes O.E. Erikss. & Winka., the results split Graphidaceae into four families: Diploschistaceae Zahlbr, Fissurinaceae B.P. Hodk, Graphidaceae s.str. and Thelotremataceae [22,23]. This classification system was adopted in the latest "Outline of Fungi and fungus-like taxa", a classification system for eukaryotic microorganisms [24].

There are more than 150 species of *Fissurina* in the world, and this genus is mainly distributed in tropical and subtropical regions [25–31]. The subtropical zone of China is south of the Qinling Mountains and the Huaihe River, north of the Leizhou Peninsula and east of the Hengduan Mountains; the tropical zone generally refers to the area between the Tropic of Cancer, such as the entire Hainan Province, the Nansha Islands, and southern Guangxi, Guangdong, Yunnan and Taiwan. The species resources are very rich in these regions and are therefore the key to revealing the biodiversity of lichens [32,33]. Until now, 17 species of *Fissurina* have been described and reported in China, all distributed in tropical and subtropical zones [7,34,35]. During a survey of the lichen diversity of southern China, a new species of *Fissurina* collected from Fujian Province, and two newly recorded species from Yunnan and Guangdong provinces were discovered and they are described in this paper; phylogenetic trees were also constructed for these species.

## 2. Materials and Methods

### 2.1. Morphological and Chemical Analyses

The Chinese materials for this study were collected from Hunan, Yunnan, Guizhou, Fujian, Guangdong and Hainan provinces, and were deposited in the Fungarium of the College of Life Sciences, Liaocheng University (LCUF) after drying and low-temperature treatment.

Morphological and anatomical features were observed and photographed using a stereo microscope (OLYMPUS SZX16) and an optical microscope (OLYMPUS BX53 with digital camera OLYMPUS DP74).

Morphological observations were made, including the growth type, color, surface state and accessory structure of the thallus, the disc and labia, and mode of initiation of the ascomata. The specific methods of anatomical research are as follows:

First, remove the well-developed ascomata, thallus and a small part of substrate as one with a blade, gently flatten the sample on white cardboard, and slice the ascomata with part of the substrate by hand with a blade. Next, select a complete, uniform, and thin slice, place it in the sterile water droplet on the glass slide, add a coverslip, absorb excess water from one side with absorbent paper, and remove air bubbles. According to the characteristics of the genus, observe the characteristics of the epithecium, hymenium, paraphyses and ascospores in detail, and take pictures and make records. This process requires a light press of the coverslip to disperse the paraphyses so that the branches at the ends of the paraphyses can be easily observed. After the ascus is exposed, the number, size and type of the ascospores in the ascus can be clearly observed. Finally, add Lugol's iodine solution dropwise on one side of the coverslip, and absorb it with absorbent paper on the other side, and test the amyloidity of the hymenium and ascospores using Lugol's solution.

Spot tests were performed on the thallus surface and thin thallus sections by adding K (10% KOH solution), C (saturated aqueous NaClO solution) and P (dissolving p-phenylenediamine in anhydrous ethanol to prepare a 5% ethanol solution) reagents. The secondary metabolites of the lichens were analyzed and identified by thin layer chromatography (TLC) with the C solvent system [36–38], specific steps are as follows:

In this study, *Lethariella cladonioides* (Nyl.) Krog. containing atranorin and norstictic acid was used as the partition standard sample.

Firstly, do the preparatory work: Prepare the solvent according to the formula (toluene: acetic acid = 200:30 mL). Then, use a 2B pencil to lightly draw a straight line about 1.5 cm from the bottom edge of the long side of the glass silicone board, (take the size

of 100 mm × 200 mm as an example), and draw a parallel straight line 0.2 cm above and below the straight line, take a point every 1 cm between the two ends of the first straight line, except the 1st, 10th and 19th, these three points are numbered as standard samples, the rest are coded as 1–16 in the sequence for sampling.

Next, sample preparation, extraction and spotting: use a blade (75% alcohol disinfection) to scrape a small amount of thallus cortex and medulla onto white paper, transfer them to sterilized small centrifuge tubes and number them; add a small amount of acetone to the small centrifuge tube until the sample is buried. After soaking for about 1–2 h, the capillary can be used to spot the sample according to the number.

Exposure layer: Pour an appropriate amount of C system solvent into the chromatography cylinder, put the glass silicone board with sample placed on it into the chromatography cylinder, and the straight line at the bottom of the glass silicone board should be above the chromatography solution. After the glass silicone board has been saturated in the chromatographic solution for 15–20 min, and before the solvent reaches 1 cm from the end of the chromatography plate, the glass silicone board can be taken out, and the surface of the plate can be dried with a hair dryer.

Color development: Observe the position and color of the chromatographic spots under sunlight; observe the fluorescence of the chromatographic spots under 365 nm and 254 nm ultraviolet light. Then, spray the silica gel plate evenly with 10% sulfuric acid solution, observe whether there are fat spots while wet, and record results. Then, put it in an oven at 85 °C for 10–15 min, so that the chromatographic color develops well.

Partition and component research: the three straight lines at the beginning of the chromatography on the silica gel plate are zone 1; draw the tangent lines on the upper and lower borders of the norstictic acid and atranorin acid spots, respectively, and define them as zone 4 and zone 7; draw a line between zone 1 and zone 4, and divide them into zone 2 and zone 3 in the middle; draw a line between zones 4 and 7 to divide them evenly into zones 5 and 6; above zone 7 is zone 8. Record the color and position of each spot, and record the Rf value if necessary.

### 2.2. DNA Extraction, Amplification and Sequencing

Genomic DNA was extracted from the ascomata and thallus of the specimens using the REDExtract-N-Amp Plant PCR Kits (Sigma-Aldrich, Saint Louis, MO, USA) according to the manufacturer's protocol. The internal transcribed spacer region (ITS), nuclear large subunit rDNA (nuLSU) and mitochondrial small subunit rDNA (mtSSU) regions were amplified using the primer pair ITS1F/ITS4 [39,40], AL2R/LR6 [41] and mrSSU1/mrSSU3R [42], respectively. The 50 μL PCR reaction system consisted of 2.5 μL each primer solution, 4 μL genomic DNA, 16 μL ddH2O and 25 μL 2×Taq PCR MasterMix (Tiangen, Beijing, China). Thermocycling conditions for ITS comprised initial denaturation at 94 °C (3 min), 35 denaturation cycles at 94 °C (30 s), annealing at 52 °C (30 s), extension at 72 °C (1.5 min) and a final extension at 72 °C (10 min); for nuLSU conditions comprised initial denaturation at 94 °C (5 min), 35 denaturation cycles at 95 °C (30 s), annealing at 58 °C (30 s), extension at 72 °C (1 min) and a final extension at 72 °C (10 min); for mtSSU conditions comprised initial denaturation at 95 °C (5 min), 35 denaturation cycles at 94 °C (45 s), annealing at 50 °C (1 min), extension at 72 °C (1.5 min) and a final extension at 72 °C (10 min). The target product of PCR was affirmed by electrophoresis on 1% agarose gels and sequenced by Biosune Inc. (Shanghai, China) and Tsingke Biotech Co., Ltd. (Beijing, China). The newly generated sequences were submitted to GenBank (Table 1). *Porina aenea* (Körb.) Zahlbr. and *P. leptalea* (Durieu & Mont.) A.L. Sm. were selected as the outgroup [21].

**Table 1.** Specimens and sequences used for phylogenetic analyses. Newly generated sequences are shown in bold.

| Species | Country | Voucher Specimens | ITS | mtSSU | nuLSU |
|---|---|---|---|---|---|
| *Clandestinotrema clandestinum* | Costa Rica | Sipman 44327 | — | JX421014 | — |
| *C. leucomelaenum* | Ecuador | Luecking 26216 | — | JX421016 | — |
| *C. leucomelaenum* | Costa Rica | Luecking LCB6b | — | JX421015 | — |
| *C. stylothecium* | Nicaragua | Luecking 28636 | — | HQ639597 | HQ639662 |
| *Cruentotrema cruentatum* | Brazil | Luecking 263 | — | HQ639587 | HQ639660 |
| *Cr. thailandicum* | Myanmar | TNS:YO12327 | LC573996 | — | — |
| *Cr. thailandicum* | Thailand | Lumbsch 19955d1 | — | JF828960 | JF828975 |
| *Cr. thailandicum* | Thailand | Lumbsch 19955d2 | — | JX421020 | — |
| *Cr. thailandicum* | Thailand | Lumbsch 19955d3 | — | JX421021 | — |
| *Dyplolabia afzelii* | Australia | MEL:2382721 | KP012902 | — | — |
| *D. afzelii* | / | RMG246 | MK503256 | — | — |
| *D. afzelii* | USA | Luecking 26509 | — | HQ639594 | HQ639628 |
| *D. afzelii* | Australia | Kalb 33915 | — | DQ431950 | AY640013 |
| *Fissurina adscribens* | China | HNX18044 | **OR264094** | **OR264108** | **OR264070** |
| *F. adscribens* | China | FJ220250 | **OR264085** | **OR264107** | **OR264068** |
| *F. adscribens* | China | FJ211195 | **OR264084** | — | — |
| *F. adscribens* | Thailand | Lumbsch 20200c | — | JX421032 | — |
| *F.* aff. *dumastii* | Thailand | Kalb 38899 | — | JX421034 | JX421487 |
| *F.* aff. *humilis* | Tanzania | A. Frisch No. 99/Tz2002 | — | DQ431948 | DQ431921 |
| *F. aggregatula* | Peru | RivasPlata 107C | — | JX421036 | JX421490 |
| *F. aggregatula* | Thailand | Luecking 24019 | — | JX421038 | — |
| *F. aggregatula* | Thailand | Kalb 38890 | — | JX421037 | — |
| *F. amazonica* | Brazil | Caceres 11030 | — | KJ608636 | KJ608633 |
| *F. astroisidiata* | Mexico | Luecking RLD057 | — | JX421040 | JX421491 |
| *F. bullata* | Australia | Mangold 6f | — | JX421041 | KF875537 |
| *F. comparimuralis* | El Salvador | Luecking 28103 | — | JX421042 | JX421492 |
| *F. crassilabra* | China | FJ211425 | **OR264081** | **OR264114** | **OR264065** |
| *F. crystallifera* | USA | Mercado 4462 | — | — | KF875538 |
| *F. dumastii* | Cameroon | Frisch & Tamnjong Idi 99/Ka3855 | — | DQ384926 | — |
| *F. illiterata* | USA | Common 9115B | — | JX421045 | — |
| *F. inabensis* | China | GZ18113 | **OR264087** | **OR264105** | **OR264071** |
| *F. inabensis* | China | YN221499 | **OR264086** | **OR264109** | **OR264067** |
| *F. inabensis* | China | FJ211545 | **OR264088** | **OR264104** | **OR264066** |
| *F. inabensis* | Russia | LE L-18645 | OP901516 | — | — |
| *F. insidiosa* | China | HN19093 | **OR264080** | **OR264111** | — |
| *F. insidiosa* | / | AFTOL-ID 1662 | — | DQ972995 | DQ973045 |
| *F. insidiosa* | USA | Spribille 39035 | KR017123 | KR017325 | KR017185 |
| *F. inspersa* | USA | Common 9113G | — | JX421047 | — |
| *F. marginata* | Australia | Kalb 33944 | — | DQ431951 | AY640012 |
| *F. marginata* | Thailand | Luecking 24122 | — | HQ639613 | JX421493 |
| *F. monilifera* | Puerto Rico | Mercado 838 | — | KJ435167 | KJ440941 |
| *F. nigrolabiata* | Philippines | Rivas Plata 1198B | — | JF828961 | JF828976 |

**Table 1.** *Cont.*

| Species | Country | Voucher Specimens | ITS | mtSSU | nuLSU |
|---|---|---|---|---|---|
| *F. nigrolabiata* | Philippines | Rivas Plata 1016E | — | — | JX421494 |
| *F. nitidescens* | USA | Common 9113C | — | JX421048 | — |
| *F. pseudostromatica* | China | YN222451 | **OR264089** | **OR264098** | **OR264074** |
| *F. pseudostromatica* | China | YN222474 | **OR264090** | **OR264099** | **OR264072** |
| *F. pseudostromatica* | USA | Common 9126B | — | JX421051 | — |
| *F. pseudostromatica* | USA | Common 9113A | — | JX421050 | — |
| *F. pseudostromatica* | Bolivia | Luecking 29014 | — | JX421049 | — |
| *F. pseudostromatica* | Thailand | Kalb 38827 | — | — | JX421495 |
| *F. pseudostromatica* | China | YN210691 | **OR264091** | **OR264100** | **OR264075** |
| *F. rufula* | Fiji | Lumbsch 20521l | — | JX421053 | JX421497 |
| *F. rufula* | Fiji | Lumbsch 20500l | — | JX421052 | — |
| *F. subcomparimuralis* | China | GD19219 | **OR264092** | **OR264112** | **OR264079** |
| *F. subcomparimuralis* | China | GD19359 | **OR264093** | **OR264113** | **OR264078** |
| *F. subfurfuracea* | Brazil | Caceres 11981 | — | KJ608635 | KJ608632 |
| *F. subundulata* | China | FJ220370 | **OR264082** | **OR264110** | — |
| *F. subundulata* | China | GZ18148 | **OR264083** | **OR264106** | **OR264069** |
| *F. triticea* | Tanzania | A. Frisch No. 99/Tz1855 | — | DQ431952 | AY640011 |
| *F. wuyinensis* | China | FJ230605 | **OR264095** | **OR264101** | **OR264076** |
| *F. wuyinensis* | China | FJ230610 | **OR264097** | **OR264103** | **OR264073** |
| *F. wuyinensis* | China | FJ230464 | **OR264096** | **OR264102** | **OR264077** |
| *Porina aenea* | Czech Republic | PRA-Vondrak25464 | OQ718026 | OQ646398 | — |
| *P. leptalea* | Czech Republic | PRA-Vondrak24457 | OQ718029 | OQ646400 | — |
| *Pycnotrema pycnoporellum* | USA | Luecking 26546 | — | JX421295 | JX421615 |

### 2.3. Phylogenetic Analysis

The contigs were assembled and edited using the program Geneious v. 9.0.2 (Biomatters Ltd., Auckland, New Zealand), and subjected to BLAST searches for an initial verification of their identities. The sequences were aligned using MAFFT v 7.308 with settings appropriate for the variability of each locus [43]. For ITS sequences, we used the L-ING-i alignment algorithm with the remaining parameters set to default values. For nuLSU, the G-ING-i algorithm and "leave gappy regions" were selected. Then, we used the E-ING-i algorithm for mtSSU with the remaining parameters set to default values. A concatenated, 3-locus matrix was generated using Geneious v. 9.0.2. This matrix contained both *Fissurina* species and species belonging to related genera in the Fissurinoideae of the Graphidaceae family [4,44]. In addition to our newly generated sequences, other related sequences were downloaded from GenBank and added to the matrix (Table 1). Maximum likelihood (ML) and Bayesian inference (BI) were performed using the CIPRES Scientific gateway portal (http://www.phylo.org/portal2/) (accessed on 12 July 2023) [45]. Maximum likelihood bootstrapping analysis was performed with RAxML HPC v. 8, using the locus-specific model partitions with the default parameters and the GTRGAMMA model as implemented on the CIPRES, NSF XSEDE resource with bootstrap statistics calculated from 1000 bootstrap replicates [46]. For the Bayesian analysis, the best substitution models of the three loci were estimated using the Akaike information criterion in jModelTest 2.1.6 [47]. Based on the results, we used the GTR+G model for ITS, SYM+I+G for nuLSU and GTR+I+G for mtSSU. Bayesian analysis was performed using MrBayes v. 3.2.2 on CIPRES with 2 independent runs, searching for 10 000 000 generations with four independent chains and sampling every 1000th tree [48]. After discarding the burn-in, the remaining 7500 trees of each run were pooled to calculate a 50% majority-rule consensus tree. Clades that received bootstrap

support ≥70% under ML and posterior probabilities ≥0.95 were considered significant. Generated phylogenetic trees were visualized under Figtree v. 1.4.2 [49].

## 3. Results and Discussion

Detailed figures of the morphology and spores, together with information regarding the chemical compositions of the three species, have been provided. Furthermore, the phylogenetic positions of these species have been confirmed.

For this study, 50 new sequences were generated (Table 1). The concatenated, three-locus matrix consisted of 65 individuals and 3253 aligned nucleotide position characters. Phylogenies derived from the ML and B/MCMC analyses were generally concordant. Minor differences in the arrangement of some terminals occurred, but relationships at deeper nodes and in well-supported clades were identical. We chose to present the ML topology, with nodal support values from both ML bootstrap analysis and posterior probabilities from the Bayesian inference (Figure 1).

The phylogenetic analysis showed that the subfamily Fissurinoideae exhibits a well-supported monophyletic lineage containing the genera *Clandestinotrema* Rivas Plata, Lücking & Lumbsch, *Cruentotrema* Rivas Plata, Papong, Lumbsch & Lücking, *Dyplolabia* A. Massal., *Fissurina* and *Pycnotrema* Rivas Plata & Lücking. The tree shows that *Fissurina* is polyphyletic in its current delimitation. The topology of our tree is similar to the results of Rivas Plata et al. and Lumbsch et al. [2,44].

The new species *Fissurina wuyinensis* formed a single clade, represented by a bootstrap support of 100, and a posterior probability of 1 for the branch (Figure 1). Its species status is further supported by its distinctive morphological, chemical and geographic characteristics (see below in Taxonomy).

*Fissurina pseudostromatica* is polyphyletic in this tree, forming a high-support clade together with *F. aggregatula*, *F. wuyinensis* and *Pycnotrema pycnoporellum* (bootstrap = 99%, posterior probability = 1). Two specimens of *F. pseudostromatica* from China were clustered with the material from USA, while another specimen was clustered with *F. aggregatula* collected from Peru. It appears to indicate issues with current species circumscriptions. Similar problems also arise in the species *F. adscribens*, *F. insidiosa* and *F. marginata*; none of them are monophyletic in our tree.

*Fissurina subcomparimuralis* from China is phylogenetically close to *F. comparimuralis* and *F. nitidescens*; they all have fissurinoid to chroodiscoid ascomata, somewhat carbonized lirellae and muriform spores, but *F. subcomparimuralis* has non-amyloid ascospores which can be distinguished from the other two species.

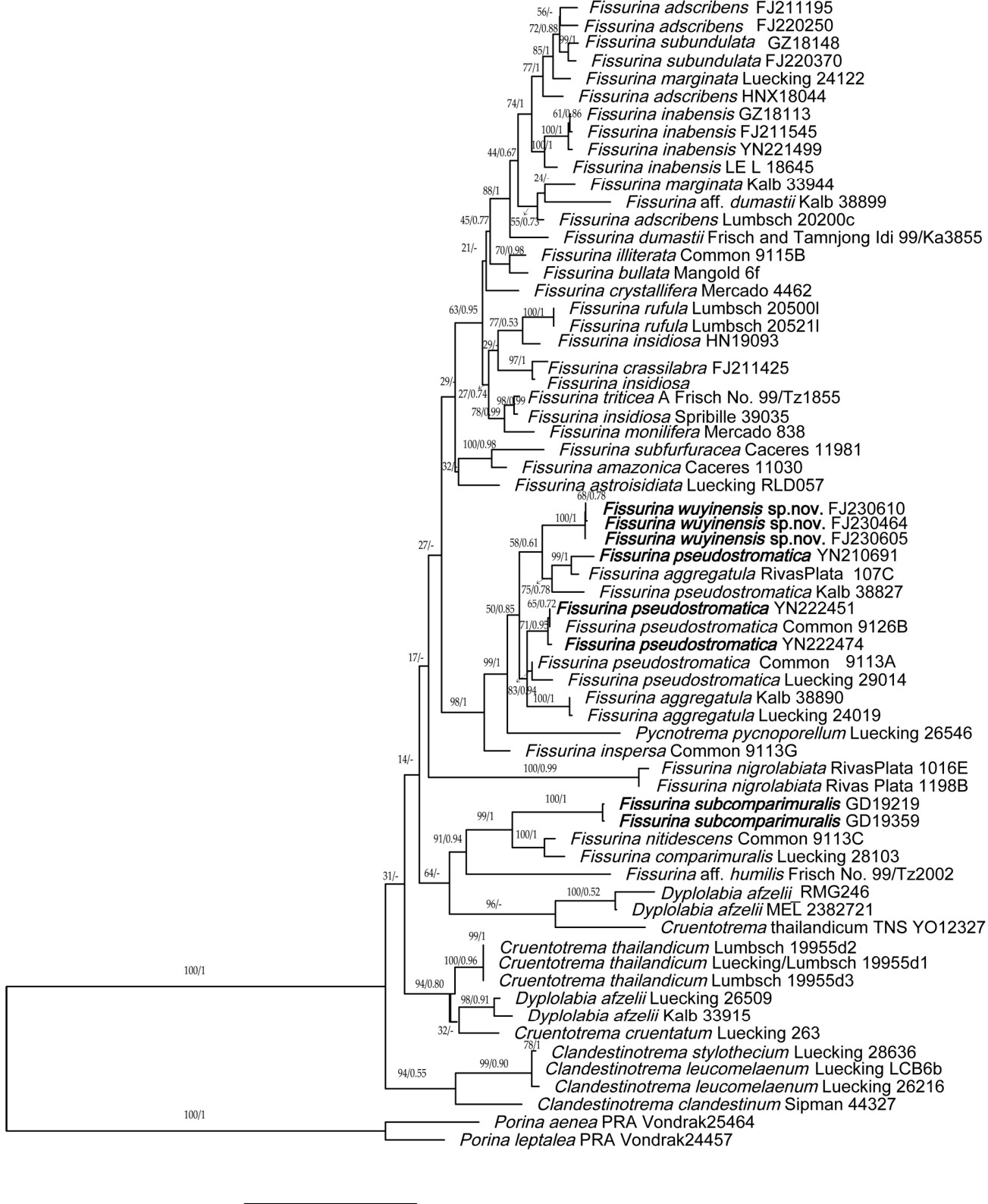

**Figure 1.** Phylogenetic tree generated from maximum likelihood (ML) analysis based on combined ITS, nuLSU and mtSSU sequences. ML bootstrap values and B/MCMC posterior probabilities (second value) are displayed above each branch. New species and records from China are shown in bold.

## 4. Taxonomy

*Fissurina wuyinensis* K.J. Shi, Z.F. Jia and X. Zhao, sp. nov., Figure 2.

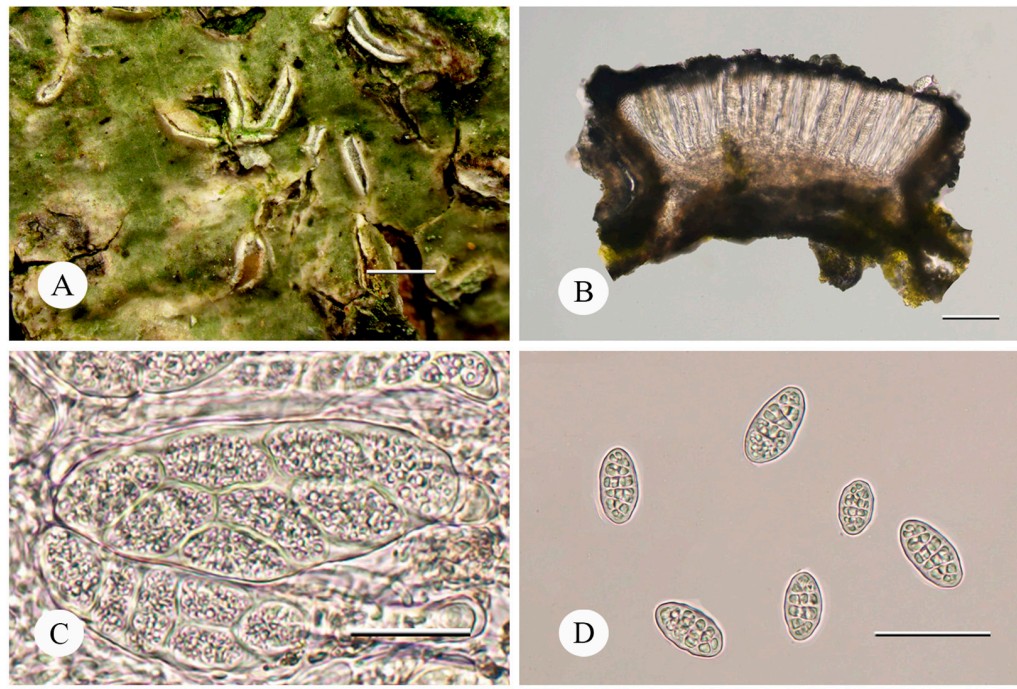

**Figure 2.** *Fissurina wuyinensis* (FJ230464) (**A**) Thallus with ascomata; (**B**) Cross-section of apothecium; (**C**) Hymenium with asci; (**D**) Ascospores. Bars: A = 1.0 mm; B = 100 μm; C = 50 μm; D = 50 μm.

**Type:** China, Fujian Province: Wuyishan City, Xingcun Town, Tongmu Village, Guadun, 27°43′52″ N, 117°39′29″ E, alt. 875 m, on bark, 9 June 2023, Jiang Shuhao FJ230464 (LCUF, holotype).

Fungal Names: FN 571620

**Description**: Thallus corticolous on tree trunk, continuous, epiperidermal, dark green to olive green, smooth, glossy. Photobiont trentepohlioid, cells angular to elongate, 6–15 × 5–9 μm. Ascomata lirellate, prominent to sessile, straight, curved or sinuous, terminally acute, single to sparsely branched, 0.8–1.5 mm long, 0.35–0.45 mm wide. Disc partially exposed and deeply immersed, labia thin but conspicuous, grey, white pruina present, thalline margin remaining inclined but with the cortex splitting off and becoming erect. Excipulum uncarbonized, yellowish brown to grayish brown, with clusters of crystals, 28–40 μm wide; epithecium black-grey, 7–16 μm wide; hymenium colorless, clear, 120–130 μm high; hypothecium brown, 25–34 μm high; paraphyses simple, apically not or sometimes slightly thickened and smooth, 0.5–0.8 μm wide; periphysoids absent. Asci broadly clavate to cylindrical, 103–138 × 24–42 μm. Ascospores 8 per ascus, ellipsoid, hyaline, muriform with 5–7 × 1–3 septa, 25–35 × 16–19 μm, without halo, I+ deep violet-blue.

**Chemistry**: thallus K–, C–, P–; no substances detected by TLC.

**Etymology**: The epithet refers to the type locality: Mount Wuyi.

**Additional specimens examined:** Fujian Province: Wuyishan City, Xingcun Town, Tongmu Village, Guadun, 27°43′52″ N, 117°39′29″ E, alt. 875 m, on the bark of *Camellia*, 9 June 2023, Guo Tangli FJ230472 (LCUF); Gaoqiao, 27°42′55″ N, 117°43′06″ E, alt 510 m, on the bark of *Camellia*, 9 June 2023, Jia Zefeng FJ230605, Jiang Shuhao FJ230610 (LCUF).

**Ecology and distribution**: The new species grows on bark at a comparatively low elevation (no more than 1000 m). It is thus far known only from the type locality.

**Notes:** The thallus and ascomata morphology and molecular sequence data place it in the genus *Fissurina* Fée [50]. In the reported *Fissurina* species of China, only three species viz. *Fissurina elaiocarpa* (A.W. Archer) A.W. Archer, *F. inabensis* (Vain.) M. Nakan. & Kashiw.

and *F. subundulata* Kalb & Z.F. Jia have muriform ascospores of similar size and lack lichen substances, but they all have a concealed disc, and *F. subundulata* and *F. inabensis* have non-amyloid ascospores [7,51].

*Fissurina wuyinensis* is characterized by uncarbonized lirellae, exposed disc with pruina, amyloid muriform ascospores and no chemistry. The new species largely resembles *Fissurina confusa* Common & Lücking, but differs in the wider lirellate (0.35–0.45 mm vs. 0.15–0.2 mm), thin-walled ascospores and in lacking lichen substances [52]. *Fissurina aperta* Herrera-Camp., Barcenas-Peña & Lücking and *F. reticulata* R. Miranda, Herrera-Camp. & Lücking have more or less similar morphological features, but the former has apically carbonized lirellae and the latter has longer ascospores about 35–45 μm [53]. In addition, it differs from *Fissurina americana* Lendemer & R.C. Harris by the prominent lirellae with exposed disc [54].

**Fissurina pseudostromatica** Lücking & Rivas Plata, Bull. Florida Mus. Nat. Hist. 49(4): 145 (2011), Figure 3.

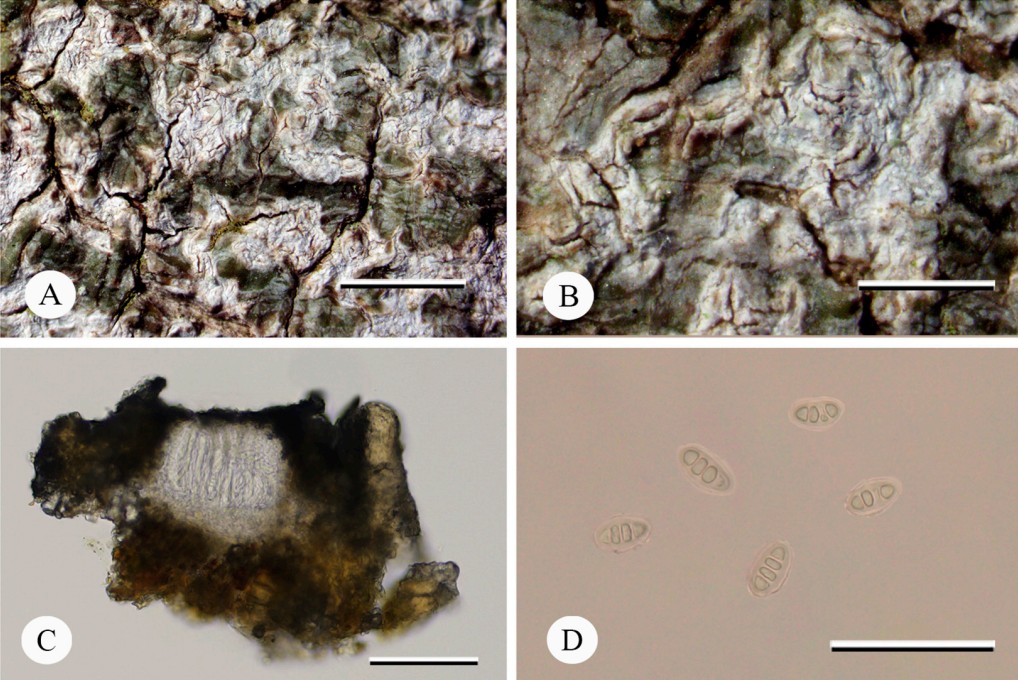

**Figure 3.** *Fissurina pseudostromatica* (YN210691) (**A**,**B**) Thallus with ascomata; (**C**) Cross-section of apothecium; (**D**) Ascospores. Bars: A = 2.0 mm; B = 1.0 mm; C = 100 μm; D = 50 μm.

**Type:** U.S.A. Florida. Collier County, Fakahatchee Strand Preserve State Park, Janes Scenic Drive 6.5 mi NNW of ranger station, west of old tram, Taxodium-Sabal hardwood hammock, slough and strand, March 2009, Lücking & Rivas Plata 26512 (F, holotype).

**Description:** Thallus corticolous, crustose, continuous, surface smooth, olive-green, glossy. Ascomata 0.1–0.4 mm long, 0.1–0.2 mm wide, lirellae densely aggregate in pseudostromatic, white clusters, short to elongate, straight to curved, single to sparsely branched, immersed to slightly raised arising. Disc concealed to slightly gaping, with thin, whitish labia, covered by concolorous thalline margin. Excipulum uncarbonized, brown to tawny, with clusters of crystals, 20–27 μm wide; epithecium grey, 3–8 μm wide; hymenium colorless, clear, 71–89 μm high; hypothecium pale brown, 10–18 μm high; paraphyses simple, 1–2.5 μm wide. Asci cylindrical, 75–83 × 7–11 μm. Ascospores 8 per ascus, ellipsoid, hyaline to pale gray-brown when old, 4 loculars, 15–20 × 7–11 μm (including 1 μm thick wall), I–.

**Chemistry:** thallus K+, C–, P–; no substances detected by TLC.

**Selected specimens examined:** China, Yunnan Province: Mengla County, Xishuangbanna Tropical Botanical Garden, Chinese Academy of Sciences, 21°55′37″ N, 101°14′51″ E,

alt. 530 m, on bark, 30 June 2021, Zhu Mengli YN210691 (LCUF); Menglun Town, Linleyuan Mountain Village, 21°55′55″ N, 101°16′09″ E, alt. 540 m, on bark, 4 September 2022, Jia Tao YN222451, YN222474 (LCUF).

**Ecology and distribution**: Growing on tree bark of the tropical rainforests at low altitudes. Previously reported from the U.S.A. and Brazil [52,55]. Newly reported for China.

**Notes:** The morphology, anatomy and chemical characteristics of the Chinese specimens are the same as those of the type specimens, but the ascospores are wider (7–11 μm vs. 5–7 μm) [52]. *Fissurina pseudostromatica* is distinguished by pseudostromatic clusters of lirellae, transversely septate and I– ascospores. *Fissurina mexicana* and *F. intercludens* have similar thallus structure, but differ in having muriform ascospores. The species is also similar to *F. aggregatula* Common & Lücking by its 4 locular ascospores and the absence of lichen substances, while the latter has labiate-aggregate lirellae and narrower ascospores (14–20 × 7–9 μm) [52,56].

*Fissurina subcomparimuralis* Common & Lücking, Phytotaxa 18: 58 (2011), Figure 4.

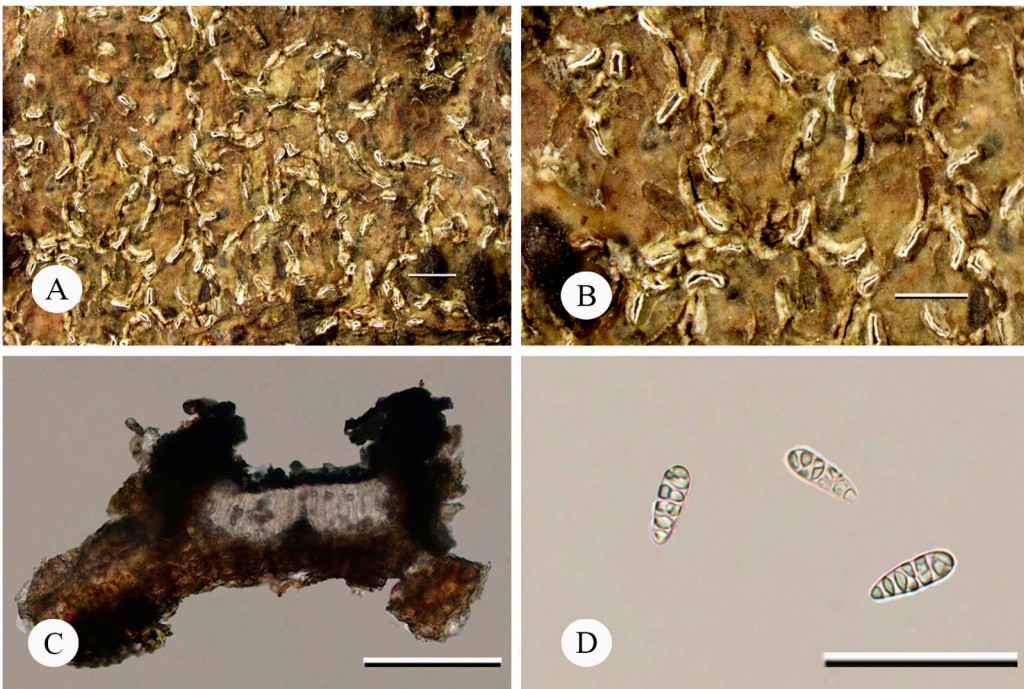

**Figure 4.** *Fissurina subcomparimuralis* (GD19219) (**A**,**B**) Thallus with ascomata. (**C**) Cross-section of apothecium; (**D**) Ascospores. Bars: A = 1.0 mm; B = 1.0 mm; C = 200 μm; D = 50 μm.

Type: U.S.A. Florida. Collier County, Fakahatchee Strand Preserve State Park, Common 7323A (holotype MSC, isotype hb. Common).

**Description**: Thallus corticolous, crustose, continuous, surface smooth, yellowish-brown to olive-brown, slightly glossy. Lirellae immersed to sessile, straight to curved, single, 0.4–1.0 mm long, 0.2–0.35 mm wide. Disc concealed to fissured; labia conspicuous, yellowish-white, covered by thalline margin; proper margin distinct, visible as thick. Excipulum entire, dark brown to brown-black, 49–61 μm wide; epithecium black-grey, 11–20 μm wide; hymenium colorless, 70–85 μm high; hypothecium grayish brown, 15–23 μm high; paraphyses unbranched, glabrous, 0.5–0.8 μm wide; periphysoids short, 10–20 μm long, indistinctly warty. Ascospores 8 per ascus, ellipsoid, colorless, muriform with 4–6 × 1–3 septa, 17–24.5 × 6–8.5 μm, weakly halonate, I–.

**Chemistry:** thallus K–, C–, P–; no substances detected by TLC.

**Selected specimens examined:** China, Guangdong Province: Guangzhou, Baiyun Mountain, Pugu, 23°10′08″ N, 113°17′32″ E, alt. 110 m, on bark, 20 January 2019, Li Min GD19219 (LCUF); South China Botanical Garden, North Entrance, 23°11′19″ N, 113°21′30″ E, alt. 22 m, on bark, 21 January 2019, Yao Zongting GD19359 (LCUF).

**Ecology and distribution:** This species grows on bark in low elevations of tropical regions. Previously reported from Florida, U.S.A. [57]. Newly reported for China.

**Notes:** The morphology and chemical characteristics of the specimens are the same as those of the type specimens, but the disc sometimes gaping and lirellae shorter. The lirellae features of GD19359 are more similar to the type specimen than GD19219 in having thin labia greyish-black to brown-black below corticate thalline margin. *F. subcomparimuralis* is similar to *F. comparimuralis* Staiger, but the latter differs in having amyloid ascospores and lacking periphysoids [14,57]. *F. subcomparimuralis* is also similar to *F. incrustans* Fée in having similar-sized, muriform ascospores and lacking lichen substances, but *F. incrustans* has I+ violet ascospores and exposed disc [7,58,59].

**Author Contributions:** Conceptualization, X.Z. and Z.J.; methodology, X.Z.; software, K.S. and X.Z.; validation, K.S., Z.J. and X.Z.; formal analysis, K.S. and X.Z.; investigation, K.S. and Z.J.; resources, Z.J.; data curation, K.S. and X.Z.; writing—original draft preparation, K.S.; writing—review and editing, X.Z.; visualization, K.S.; supervision X.Z.; project administration, Z.J. and X.Z.; funding acquisition, Z.J. and X.Z. All authors have read and agreed to the published version of the manuscript.

**Funding:** This research was funded by the Emergency Management Project of the National Natural Science Foundation of China, grant number 31750001, the Young Scientists Fund of National Natural Science Foundation of China, grant number 31700018 and Shandong Provincial Natural Science Foundation, China (ZR2023MC105).

**Institutional Review Board Statement:** Not applicable.

**Data Availability Statement:** Publicly available datasets were analyzed in this study. This data can be found from http://www.ncbi.nlm.nih.gov/ (accessed on 14 July 2023).

**Acknowledgments:** We are grateful to Jiang Shuhao and Guo Tangli for their help with field research.

**Conflicts of Interest:** The authors declare no conflict of interest.

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
