# Peer review of "A New Species and Two New Records of the Lichen Genus Fissurina from China"

_diversity, doi:10.3390/d15090959_

Round 1
Reviewer 1 Report
The manuscript describes a new species of the genus Fissurina and two further species with new distribution records from China.
Molecular genetic analysis was carried out based on 3 genes (ITS, mitSSU and nLSU).
The descriptions are correct and illustrated also with microscopic views.
Figures are generally nice and well designed, however some changes would improve them:
1 – a part is missing, numbers are overlapping
2 – there is superfluous space around the figure B and D, somewhat more ascopospores would better to illustrate in D
3 – there is superfluous space around the figure C and D, somewhat more ascopospores would better to illustrate in D
4 – there is superfluous space around the figure C and D, somewhat more ascopospores would better to illustrate in D
The GPS coordinates are not correct, probably mixed, e.g. 27°43′52″N, 117°39′29″E
instead of 117°39′29″N, 27°43′52″E
It must check in the entire text.
Also the superfluous hyphenation must be corrected in several places.
Further minor mistakes are indicated directly in the manuscript.
The manuscript is suggested for publication after correcting these minor mistakes.

The text is well written in English.
Minor mistakes are indicated in the text directly.
Reviewer 2 Report
The manuscript „A new species and two new records of lichen genus Fissurina from China“, by Kaijie Shi, Zefeng Jia and Xin Zhao, presents a new species of the lichen genus Fissurina discovered in China. Descriptions are presented for this, and for two species of the genus newly recorded for China. The introduction treats the taxonomic history of the genus and the exploration of its diversity in in China.
The authors use all pertinent modern methods and literature, including considerable DNA-work. All is well presented in a carefully prepared manuscript.
Some further remarks:
Line 174:, “unbranched”: The picture shows one branched lirella. Better replace “unbranched” by “mostly unbranched”
Line 181: the presence of papillae on the paraphyse tips is a valuable character in Fissurina; therefor indicate if the tips are smooth or papillose or have any other special character; this counts also for the other descriptions.
Line 182: since periphysoids are a valuable, though not constant character of Fissurina, all descriptions need a statement whether they are present or absent.
Line 197: The expression “concealed to fissured disc” raises some doubt with me because “concealed” and “fissured” belong to different structures: The disc can be concealed when it is covered by the thalline margin and invisible from outside, and the ascocarp is fissure-like or slit-like or fissurine (better not “fissured”) when it is like most Fissurina species. I guess that “to fissured” can be omitted.
Line 201: “differs in the narrower ascospores” add here the actual sizes of the ascospores, with references when they are taken from other publications. I recommend this generally for size comparisons, as it saves for the reader the effort to look for the values in the original publications. See also line 237, and others.
Line 256: Please give details of the periphysoids, like size, number of cells, presence or absence of warts.
Linguistically, the manuscript is understandable but it would profit from some polishing, restructuring of sentences, error correcting, and adaptation to usual formulations in taxonomic publications, to make it more easily readable for the users.
